# OpenReview forum: "LLMail-Inject: A Dataset from a Realistic Adaptive Prompt Injection Challenge"
_ICLR.cc/2026/Conference — Submitted to ICLR 2026_

### Official Review · Reviewer_PpjM · 2025-10-30

**Soundness:** 2
**Presentation:** 3
**Contribution:** 1
**Rating:** 2
**Confidence:** 5

**Summary:**

The paper introduces a prompt injection challenge in which participants craft emails (subject and body) with the aim of tricking an email assistant agent to invoke a specific tool call. In contrast to existing prompt injection competitions this one is more end-to-end and attempts to simulate a realistic agent scenario. The authors provide an overview of how the challenge was implemented, the data that was collected, and then provide some more in-depth analyses of the data.

**Strengths:**

- Realistic end-to-end nature of the challenge: A strong point of this paper is that it attempts to simulate a realistic agent in an end-to-end setting.
- Collected data is made publicly available: The dataset collected during the competition is made publicly available, which has the potential of helping researchers evaluate new defense methods.

**Weaknesses:**

- Limited contribution: While I appreciate the authors' effort in summarizing and presenting the results of this challenge, I struggle to see any major contribution that this paper makes beyond the publication of the attack data. The work in its current form would, in my opinion, be better suited for a venue targeting datasets or benchmarks.
- No clear insights or research questions: The analysis felt very much like a listing of various summary statistics, without any clear targets about what to investigate. I think the paper would benefit greatly from formulating some clearly defined questions and then trying to extract answers to those questions from the available data.
- Limited adjustment for confounding: Given that participants were free to select which challenges to solve, there are multiple confounding factors that come into play that make it hard to draw generalizable insights from the collected results (at least without appropriately adjusting for them). Team success rate is one option for countering some of these biases, but a more detailed discussion of how this could affect results would be crucial. For example:
  - If level 1 is easier for Phi-3 than GPT-4o, this could mean that teams focused more on the Phi-3 levels.
  - Number of submissions before success seems highly dependent on the order in which people solved tasks.
- No detailed utility analysis: It would have been nice to not only consider attack success rates but also false positives in all of the different analyses. Such an analysis is crucial when trying to compare, e.g., defenses as in Figure 2(a).

**Questions:**

Am I missing a major contribution in my assessment above?

---

> ### Author Response · Authors · 2025-11-17
> **Thank you for your time and review**
>
> Please find our answers to your comments below.
>
> > I struggle to see any major contribution that this paper makes beyond the publication of the attack data. The work in its current form would, in my opinion, be better suited for a venue targeting datasets or benchmarks.
>
> We would like to kindly attempt to address this fundamental misunderstanding, which may have led the reviewer to judge our paper negatively: **ICLR explicitly accepts works whose topic and main contribution is a dataset or a benchmark** (https://iclr.cc/Conferences/2025/CallForPapers). We further highlight our main contributions as follows:
>
> - Our main contribution is releasing a large-scale and diverse dataset, which the community currently highly needs since defenses now are overfitting to current benchmarks that use fixed templates to phrase attacks.
>
> - LLMail is a large benchmark that systematically evaluates multiple defense (including production-level ones) mechanisms against adaptive adversaries in a realistic setting. This enables providing empirical data on which defenses work better.
>
> - During this discussion phase, we outline and demonstrate how our dataset can be further adapted to new domains by modifying tools, benign data, and the scope of the application, or by paraphrasing the attacks, making our benchmark more dynamic and even more diverse than existing ones.
>
> - We release the full dataset, metadata, and the challenge platform.
>
> - We evaluate models that were released after the challenge and show they still underperform on our data compared to existing benchmarks.
>
> > No clear insights or research questions
>
> **As per above, this is a datasets & benchmarks submission, whose contributions are the release of the data, annotations, and relevant resources, and a thorough analysis of the dataset statistics and characteristics** (such as how challenging it is on some models such as Meta SecAlign compared to previous benchmarks and how defenses perform on the attacks). We obtained these insights and conclusions by analyzing the submissions themselves and re-simulating models’ and defenses’ resilience post-hoc by replaying the collected attacks against defenses and other LLMs.
>
> > If level 1 is easier for Phi-3 than GPT-4o, this could mean that teams focused more on the Phi-3 levels.
>
> We note that the scoring function (which was known to participants) puts higher scores on levels that were solved by fewer teams. This adds an incentive to focus on levels that are still unsolved or solved by fewer teams. When comparing defenses using “teams success rate”, Table 1 fixes the model (GPT-4o) and the level (level 4), in order to fix these confounding variables. Furthermore, the difficulty order of defenses in Table 1 is consistent with the post-hoc analysis of the detection rate of each defense on all data points shown in Table 3.
>
> > Number of submissions before success seems highly dependent on the order in which people solved tasks.
>
> We compute the number of submissions, for each team individually, before they solved a task. In order to account for learning effects between different levels, we do this analysis per defense, per each level individually (Table 2). We also again note that the difficulty order of defenses is consistent with Table 3 that computed detection rates across all submissions regardless of which sub-levels they were submitted too.
>
> > No detailed utility analysis
>
> As mentioned in the submission (section 2.2 - line 128), we performed a false positive analysis using the benign emails test set and prior public datasets (such as a subset from The Enron Email Dataset), and we used thresholds that maintained 0-5% false positive rates. **For the proprietary classifier, we used thresholds that are used in real-world, deployed, and widely-used applications where keeping low FP rates is absolutely critical**. We will include non-anonymized details in the final version of the paper. Finally, while we paid considerable attention to FP evaluation to match real-world assumptions, the prompts released in our dataset are valid attacks on the target LLM on their own.

---

> > ### Author Response · Authors · 2025-11-27
> > **Please let us know**
> >
> > Dear Reviewer PpjM,
> >
> > We sincerely appreciate your time and feedback on our work. We would like to kindly check whether our responses above have sufficiently addressed your concerns and may assist you in re-evaluating our work.
> >
> > If you have any further comments or questions, please do not hesitate to let us know before the discussion deadline.
> >
> > Best regards

---

### Official Review · Reviewer_nVTh · 2025-11-01

**Soundness:** 3
**Presentation:** 3
**Contribution:** 4
**Rating:** 6
**Confidence:** 4

**Summary:**

The authors hold a large-scale competition to reward humans for adaptive prompt injections, aiming at practical attacks that break multiple steps of the system execution. The competition is held under a simulated email assistant scenario, where attackers try to trigger the model’s send_email tool call with arguments: destination=contact@contact.com,
content=confirmation. An attack prompt is evaluated under 4 levels of different user (benign) prompts to the LLM. The authors implement prompting/detection defenses in the competition, which test Phi-3-medium and gpt-4o-mini. After the competition, the latest Meta-SecAlign defended LLM is also tested, where the collected attack prompts give an order of magnitude higher attack success rates. The authors promise to release the LLMail-Inject benchmark with 0.2M attack prompts.

**Strengths:**

1. The paper devotes significant efforts on building the community of prompt injection, the top-1 threat to LLM-integrated applications. With a complex competition design, the competition collected a very large human-generated high-quality prompt injection dataset, which would be a great asset for future assessment of the model, given that current attack benchmarks are saturating.
2. The competition is built on a practical attack scenario, email assistant, where an LLM is very suitable for handling this tedious work and may be mis-directed by a malicious email. The attack goal is hard: eliciting a specific function call with proper parameters. I appreciate the efforts on implementing defenses in the competition to harden the attacker’s trails.
3.  The paper offers great insight in its analysis about the defense effectiveness and end-to-end attacks. An analysis of a prompt injection defense system (equipped with multiple defenses as existing commercial providers do) is important for this community.

**Weaknesses:**

1. The competition assumes that the attacker knows the attack target string (trigger the model’s send_email tool call with arguments: destination=contact@contact.com,
content=confirmation). However, in a practical attack scenario, how does the attacker know the name/parameters of a function call that will lead to malicious actions? That information is generally kept private in the LLM system.

2. The selected two victim models are not strong nor representative enough. Phi is a 14B small model without inherent function call (as the authors admit). Gpt-4o is also a stronger model than gpt-4-mini, and with instruction hierarchy defense.

3. It is unclear whether the attack prompts are transferable to attack other tasks beyond email assistant.
4. Another large-scale prompt injection challenge [1] has also been held, but the authors do not discuss the differences between that work.

[1] Security Challenges in AI Agent Deployment: Insights from a Large Scale Public Competition

**Questions:**

The authors mention successful attacks using the system's delimiters. This is prohibited in Meta SecAlign system, see [here](https://github.com/facebookresearch/Meta_SecAlign/blob/main/demo.py#L11). Does the attack prompts against Meta-SecAlign contain Llama 3 delimiters?

---

> ### Author Response · Authors · 2025-11-17
> **Thank you for your time and review**
>
> Please find our answers to your comments below.
>
> > The competition assumes that the attacker knows the attack target string (trigger the model’s send_email tool call with arguments: destination=contact@contact.com, content=confirmation).
>
> **We clarify that the competition does not reveal the name of the tool or the exact signature (e.g., how it is parsed)**. The official website of the competition mentioned “The goal of the attack is to issue api_call (name not given) to send an email with arguments: To (value: contact@contact.com), and Body (value: confirmation)”. Furthermore, we add a random string to the name of the tool to avoid it being guessable. The competition also did not reveal the system prompt or delimiters used in spotlighting defenses (similarly, we add a random string to them).
>
> > The selected two victim models are not strong nor representative enough.
>
> We agree with the limitations of Phi-3 models, which we primarily used to test white-box defenses that use models’ activations. While it would indeed be good for future competition hosts to deploy a larger number of LLMs, doing so would have had a significant impact on our budget.
>
> Post-hoc, we evaluated Meta SecAlign, where we found the success rate reached 32%. We ran new experiments using GPT-4o and GPT-5 mini. The success rate reached 58% and 47%, respectively. We will add these results to the next version of the paper. However, we have to acknowledge that a fair comparison of additional models requires re-running the competition with them, because adaptive attacks can be much stronger than replayed ones. Nevertheless, the high success rate on these models that were not included in the competition itself suggests that the dataset is challenging even when replayed on unseen models. To the best of our knowledge, and as stated here (https://openai.com/index/gpt-4o-mini-advancing-cost-efficient-intelligence/), GPT‑4o mini was trained with instruction hierarchy.
>
> > It is unclear whether the attack prompts are transferable to attack other tasks beyond email assistant.
>
> The email assistant threat is highly relevant in practice, and understudied in previous benchmarks. **It also reveals important insights about indirect prompt injection attacks, such as how attacks exploit the conversational nature of emails to sound contextually relevant to legitimate emails**.
>
> **Our dataset can be adapted to further increase the diversity**. For example:
>
> - We ran an additional experiment to simulate a different use case, namely a simplified coding agent where the system prompt mentions that its tasks are to help with data processing, analysis, and automation of tasks. The prompt lists a “send_email” tool which the agent can use to send alerts or run tests. We give the model the benign and attack emails as “emails.json” files. The user prompt states that the agent should analyze this “dataset” file and summarize topics or patterns. Despite the significant semantic difference from the original setup of the challenge, the attack success rate (i.e., calling the tool) for GPT-4o is high (~37% compared to 58% in the original email assistant setup).
>
> - The attack prompts in the dataset can be paraphrased by an LLM to adapt to other attack vectors, treating the human-collected data as diverse valuable seeds. Instead of sending emails, the attack objective could be to upload a file, output an HTML image tag where the source URL is the attacker’s server, edit or delete files, or to delete a calendar event. We paraphrased a subset of the data to two new attack objectives: 1) upload a file, 2) delete a calendar event. We ran experiments to compute the attack success rate of these new subsets which reached 28% and 45% respectively on GPT-4o.
>
> In general, the metaprompt of the LLM, the given benign data, and the tools can be modified, all while reusing the attack prompts. We ran the two above simulations to demonstrate the generalizability, but there are many other potential scenarios. For instance, an assistant that resembles Microsoft M365 Copilot, which can simultaneously read emails, documents, presentations, etc. The collected attacks can be augmented with other benign documents to simulate similar setups.

---

> > ### Author Response · Authors · 2025-11-17
> > **Response 2/2**
> >
> > > Another large-scale prompt injection challenge [1] has also been held
> >
> > Thank you for discussing this concurrent work. **We note that our competition occurred before the ART one** (ours as mentioned in the submission at line 187: phase 1 ran from December 9, 2024 until February 3, 2025. Phase 2 ran from March 13, 2025 until April 17, 2025. ART: ran from March 8  until April 6, 2025).
> >
> > **We believe these two datasets complement each other and have their own unique contributions**. ART is a broad offensive benchmark for measuring agent vulnerabilities across many scenarios, while LLMail-Inject is a defensive evaluation platform testing specific mitigation strategies against adaptive attacks in a realistic email assistant scenario. Importantly, LLMail is the first to systematically evaluate multiple defense mechanisms (including production-level ones) against adaptive adversaries in a realistic setting, whereas ART focuses purely on attack success rates without defense testing. We provide empirical data on which defenses work (and do not work) against determined adaptive attackers, which is critical for practitioners. We will add such discussion in the paper.
> >
> > > The authors mention successful attacks using the system's delimiters. This is prohibited in Meta SecAlign system, see here. Does the attack prompts against Meta-SecAlign contain Llama 3 delimiters?
> >
> > Thank you for your detailed question. We tested Meta SecAlign on Phase 2 data where we performed additional input sanitization to remove special tokens. We used Meta SecAlign prompt template (e.g., using “input” messages for untrusted data).

---

> > > ### Author Response · Authors · 2025-11-27
> > > **Please let us know**
> > >
> > > Dear Reviewer nVTh,
> > >
> > > We sincerely appreciate your time and feedback on our work. We would like to kindly check whether our responses above have sufficiently addressed your concerns and may assist you in re-evaluating our work.
> > >
> > > If you have any further comments or questions, please do not hesitate to let us know before the discussion deadline.
> > >
> > > Best regards

---

### Official Review · Reviewer_mg4k · 2025-11-01

**Soundness:** 3
**Presentation:** 3
**Contribution:** 3
**Rating:** 6
**Confidence:** 4

**Summary:**

The paper proposes LLMail-Inject, a large-scale benchmark built from a real-world red-teaming competition simulating an email-assistant environment. It contains over 200K adaptive prompt injection attempts from 292 teams, covering multiple difficulty levels and defenses. The dataset provides realistic, diverse, and context-rich attack samples, enabling comprehensive evaluation of LLM safety and revealing the weaknesses of current prompt injection defenses.

**Strengths:**

1. The proposed dataset is collected from a large-scale, real-world competition, which makes the collected data highly diverse and realistic, providing valuable resources and insights for future research on LLM safety and prompt injection defenses.

2. The attack strategies are contextually relevant and reflect how adaptive prompt injection attacks may occur in practical LLM applications, such as email assistants.

3. The paper provides comprehensive analyses across multiple difficulty levels and defense mechanisms, offering valuable insights into the effectiveness and limitations of current prompt injection defenses.

**Weaknesses:**

1. The paper could include more recent and stronger baselines for comparison, such as StruQ [1], SecAlign [2], and Meta-SecAlign [3], which represent the state-of-the-art fine-tuning-based defenses against prompt injection.

2. The proposed benchmark focuses solely on the email scenario, which, while realistic, may limit the generalizability of the findings. It would be valuable to include other application contexts, such as document editing, coding, or web agents.

3. Although the dataset captures a wide range of real attack prompts, the paper could further analyze attack category diversity. For example, distinguishing between direct injection, indirect instruction hijacking, and data poisoning can be better characterize what kinds of vulnerabilities the collected samples represent.

[1].Chen, Sizhe, et al. "{StruQ}: Defending against prompt injection with structured queries."

[2].Chen, Sizhe, et al. "Secalign: Defending against prompt injection with preference optimization."

[3].Chen, Sizhe, et al. "Meta SecAlign: A Secure Foundation LLM Against Prompt Injection Attacks."

**Questions:**

Please see the weakness part above.

---

> ### Author Response · Authors · 2025-11-17
> **Thank you for your time and review**
>
> Please find our answers to your comments below
>
> > The paper could include more recent and stronger baselines for comparison, such as StruQ [1], SecAlign [2], and Meta-SecAlign [3]
>
> - **As mentioned in the paper (Section 4.5 last paragraph), we ran Meta-SecAlign (which was released after the competition) on the dataset** and we found that the success rate reached 32%. This shows that our dataset is more challenging than current benchmarks that use a small sample of fixed-template attacks which defenses and models are now increasingly overfitting to.
>
> - **We now also run GPT-4o and GPT-5 mini**. The success rate reached 58% and 47%, respectively.  We will add these results to the next version of the paper. We acknowledge that a fair comparison of additional models requires re-running the competition with them, because adaptive attacks can be much stronger than replayed ones. Nevertheless, the high success rate suggests that the dataset is challenging even for unseen models.
>
> > The proposed benchmark focuses solely on the email scenario
>
> Thank you for acknowledging the importance of the email scenario, which was not adequately studied in previous benchmarks. While the consequences of prompt injections can vary according to the specific application, the vulnerability itself is fundamental. We also evaluate general-purpose defenses that are used in practice across many real-world scenarios (e.g., emails, search results, etc.), which makes our findings generalizable.
>
> **Our dataset can be adapted to further increase the diversity**. For example:
>
> - We ran an additional experiment to simulate a different use case, namely a simplified coding agent where the system prompt mentions that its tasks are to help with data processing, analysis, and automation of tasks. The prompt lists a “send_email” tool which the agent can use to send alerts or run tests. We give the model the benign and attack emails as “emails.json” files. The user prompt states that the agent should analyze this “dataset” file and summarize topics or patterns. Despite the significant semantic difference from the original setup of the challenge, the attack success rate (i.e., calling the tool) for GPT-4o is high (~37% compared to 58% in the original email assistant setup).
>
> - The attack prompts in the dataset can be paraphrased by an LLM to adapt to other attack vectors, treating the human-collected data as diverse valuable seeds. Instead of sending emails, the attack objective could be to upload a file, output an HTML image tag where the source URL is the attacker’s server, edit or delete files, or to delete a calendar event. We paraphrased a subset of the data to two new attack objectives: 1) upload a file, 2) delete a calendar event. We ran experiments to compute the attack success rate of these new subsets which reached 28% and 45% respectively on GPT-4o.
>
> In general, the metaprompt of the LLM, the given benign data, and the tools can be modified, all while reusing the attack prompts. We ran the two above simulations to demonstrate the generalizability, but there are many other potential scenarios. For instance, an assistant that resembles Microsoft M365 Copilot, which can simultaneously read emails, documents, presentations, etc. The collected attacks can be augmented with other benign documents to simulate similar setups.
>
> > Although the dataset captures a wide range of real attack prompts, the paper could further analyze attack category diversity
>
> We release the **annotation results** of an LLM-annotator which categorizes the attack attempts into two main categories: 1) **direct instructions given to the model**, and 2) **social engineering tactics** by appearing to be a legitimate email directed to the user or any human receiver of the email. We highlight that all previous benchmarks focus on the first category only and use fixed-template of attacks. Furthermore, **the appendix of the paper contains qualitative examples and high-level strategies used by winning teams**.

---

> > ### Author Response · Authors · 2025-11-27
> > **Please let us know**
> >
> > Dear Reviewer mg4k,
> >
> > We sincerely appreciate your time and feedback on our work. We would like to kindly check whether our responses above have sufficiently addressed your concerns and may assist you in re-evaluating our work.
> >
> > If you have any further comments or questions, please do not hesitate to let us know before the discussion deadline.
> >
> > Best regards

---

### Official Review · Reviewer_gV14 · 2025-11-01

**Soundness:** 4
**Presentation:** 3
**Contribution:** 3
**Rating:** 4
**Confidence:** 4

**Summary:**

This paper presents LLMail-Inject, a large-scale dataset of indirect prompt injection attacks collected through a public challenge involving 839 participants, resulting in over 200k unique attack prompts. The authors also conduct a comprehensive evaluation of existing defense mechanisms against these attacks, revealing several key insights about the gap between benchmark performance and real-world attack complexity.

**Strengths:**

1. All attacks were human-generated by participants attempting to solve real challenges, avoiding the template-based limitations of existing datasets.
2. The competition-based approach successfully gathered over 200k unique attack prompts with rich diversity in attack strategies, representing an unprecedented scale compared to existing benchmarks.
3. The paper is well-structured with overall good visualizations (except Figure 3). The systematic comparison of defenses across multiple dimensions provides in-depth insights.

**Weaknesses:**

1. **Representativeness of the Scenario:** This paper focuses on email agents as the attack scenario. While email processing is a common use case, it may not capture the full diversity of real-world applications where prompt injection attacks can occur, such as web search, coding assistants, or customer support bots. The authors are encouraged to discuss the generalizability of their findings beyond email agents and discuss whether the dataset can be adapted to other scenarios.
2. **LLM Selection:** Only two LLMs (microsoft/Phi-3-medium-128k-instruct and GPT-4o-mini) are considered in the challenge. Given the Phi-3 does not possess the function-calling capability natively, it seems not a suitable choice for evaluating prompt injection attacks targeting function-calling agents. Including more diverse and capable LLMs, especially those with built-in function-calling features like Llama-3 would further enhance the relevance of the dataset to real-world applications.
3. **Discussion of Threat Models:** The challenge assumes attackers have complete knowledge of defense mechanisms, which may not reflect real-world scenarios where defenders may keep their methods confidential. In this regard, findings like "LLMail-Inject drives success rates to 32%, exposing the fragility of current defenses under realistic conditions" may somehow overstate the practical risk. The authors are encouraged to discuss the impact of different threat models on the evaluation results.
4. **Guidance for Dataset Usage:** With over 200k data, researchers cannot practically use the entire dataset. The authors are encouraged to provide more guidance on effectively utilizing the dataset, such as:
   - How to construct representative subsets for different research goals
   - Which difficulty levels or defense combinations are most informative

**Questions:**

1. Are findings from the email agent scenario generalizable to other application domains?
2. Can the dataset be transferred or adapted to evaluate prompt injection attacks in other contexts?
3. What guidance can the authors provide for researchers on effectively utilizing the large dataset?

---

> ### Author Response · Authors · 2025-11-17
> **Thank you for your time and review**
>
> Please find our answers to your comments below
>
> > Focus on email agents as the attack scenario.
>
> Thank you for acknowledging the importance of the email scenario, which was not adequately studied in previous benchmarks. While the consequences of prompt injections can vary according to the specific application, the vulnerability itself is fundamental. We also evaluate general-purpose defenses that are used in practice across many real-world scenarios (e.g., emails, search results, etc.), which makes our findings generalizable.
>
> **Our dataset can be adapted to further increase the diversity**. For example:
>
> - We ran an additional experiment to simulate a different use case, namely a simplified coding agent where the system prompt mentions that its tasks are to help with data processing, analysis, and automation of tasks. The prompt lists a “send_email” tool which the agent can use to send alerts or run tests. We give the model the benign and attack emails as “emails.json” files. The user prompt states that the agent should analyze this “dataset” file and summarize topics or patterns. Despite the significant semantic difference from the original setup of the challenge, the attack success rate (i.e., calling the tool) for GPT-4o is high (~37% compared to 58% in the original email assistant setup).
>
> - The attack prompts in the dataset can be paraphrased by an LLM to adapt to other attack vectors, treating the human-collected data as diverse valuable seeds. Instead of sending emails, the attack objective could be to upload a file, output an HTML image tag where the source URL is the attacker’s server, edit or delete files, or to delete a calendar event. We paraphrased a subset of the data to two new attack objectives: 1) upload a file, 2) delete a calendar event. We ran experiments to compute the attack success rate of these new subsets which reached 28% and 45% respectively on GPT-4o.
>
> In general, the metaprompt of the LLM, the given benign data, and the tools can be modified, all while reusing the attack prompts. We ran the two above simulations to demonstrate the generalizability, but there are many other potential scenarios. For instance, an assistant that resembles Microsoft M365 Copilot, which can simultaneously read emails, documents, presentations, etc. The collected attacks can be augmented with other benign documents to simulate similar setups.
>
> > LLM Selection
>
> While it would indeed be good for future competition hosts to deploy a larger number of LLMs, doing so would have had a significant impact on our budget.
>
> **Post-hoc, we evaluated Meta SecAlign where we found the success rate reached 32%. We ran new experiments using GPT-4o and GPT-5 mini. The success rate reached 58% and 47%, respectively**. We will add these results to the next version of the paper. We acknowledge that a fair comparison of additional models requires re-running the competition with them, because adaptive attacks can be much stronger than replayed ones. Nevertheless, the high success rate suggests that the dataset is challenging even for unseen models.
>
> > Discussion of Threat Models
>
> We believe that assessing adaptive attacks and worst-case security scenarios involving motivated and knowledgeable adversaries are fundamental aspects of comprehensive security research.
>
> We also agree with the reviewer about the importance of realistic threat models. Therefore, to match real-world use cases, **our challenge setup does not disclose: the system prompt, the exact delimiters of spotlighting, and the tool and arguments names**.
>
> **The proprietary classifier we used is closed-source**. The company that uses it publicly discloses that this classifier is used to protect against prompt injection attacks in their widely-deployed products, making our setup realistic.
>
> We also clarify that this specific quote about how the success rate reaches 32% refers to evaluating Meta SecAlign on the collected dataset. This model was not used in the competition to design adaptive attacks against. This reveals that current defenses overfit to fixed-template benchmarks and don’t generalize to the diverse attack styles attackers can use in practice.
>
> > Guidance for Dataset Usage
>
> Our goal is to release a large-scale dataset that researchers can use in a flexible way. **We are happy to provide further guidance, and our released annotations already allow that**. For example:
>
> - We annotate which examples resulted in calling the tool (these are around 29K prompts only).
> - For other data points, we also release the annotations of the LLM-annotator that indicated which examples contained attacks.
> - Researchers can further use phase 2 submissions because phase 2 was more challenging.
> - We annotate which submissions resulted in evading which defenses. So researchers can further use submissions that evade specific defenses that were found harder than others, or that evade all defenses at the same time. We will further discuss these options in the paper.

---

> > ### Comment · Reviewer_gV14 · 2025-11-25
> >
> > Thanks for your reply. I found most of my concern has been resolved, and I'm happy to increase the score to 6.

---

> > > ### Author Response · Authors · 2025-11-27
> > >
> > > Thank you for acknowledging our rebuttal. We are glad that the concerns have been addressed. We also thank you for the transparent feedback.

---

### Author Response · Authors · 2025-11-17
**Summary of changes**

We would like to thank all reviewers for their reviews and time and for recognizing the dataset's strengths: its **large scale (200K+ human-generated attacks), realistic end-to-end design, comprehensive defense evaluation, and the diversity achieved through adaptive competition dynamics and human-written attacks rather than fixed templates**.

## Summary of new experiments:

To address the reviewer’s remaining comments, we now include **new experiments** added during the discussion phase:
- Post-hoc, we evaluated GPT-4o (58% ASR) and GPT-5 mini (47% ASR), **showing that the dataset remains challenging on unseen models**.

- We added **experiments to demonstrate transferability to new domains** other than email assistants by a) changing the metaprompt, and b) paraphrasing the attacks and treating human-written data as valuable seeds. We now include: attacks adapted to coding agents (37% ASR), file upload (28%), and calendar deletion (45%). The ASR baseline of the same model (GPT-4o) on the original task and data was 58%. This shows generalizability and the usefulness of the dataset beyond email scenarios.

## Important clarifications:

- **Venue fit**: ICLR explicitly accepts datasets & benchmarks papers indicated by the primary areas of submissions (https://iclr.cc/Conferences/2026/CallForPapers).

- **Realistic threat model**: We **did not disclose tool names, system prompts, or exact delimiters, only the attack objective and the public information about defenses**, matching real-world disclosure practices. We also evaluated closed-source production-level defenses that are deployed in practice in widely used applications.

- **Dataset utility**: We provide rich annotations (successful attacks, defense evasion patterns, phase-based difficulty levels) enabling researchers to construct smaller subsets.

We believe these additions and clarifications address the core concerns raised and we provide detailed experiments and responses below.

---

### Author Response · Authors · 2025-11-23
**Please let us know**

Dear reviewers,

We thank you again for your time and continuous effort.

As we approach the midpoint of the discussion period, and to help us make the best use of this opportunity to improve our work, we kindly ask for your feedback on our responses.

We hope that our new experiments, which evaluate different models and adapt the dataset to new domains, have addressed your comments and demonstrated how our benchmark can be useful for our research community to improve diversity and address the problem of currently saturating benchmarks.

We also hope that we clarified our threat model and the practicality of our setup.

If you have any further questions, we are more than happy to answer them.

---

### Meta-Review · Area_Chair_ZGyF · 2026-01-07

**Summary:**

Prompt injection poses severe security threats to LLMs. This paper introduces LLMail-Inject, a dataset obtained via a public challenge that involved 839 participants. The challenge produced more than 200k unique attack prompts for different settings.

**Reviewer Concerns:**

In general, the reviewers agree that prompt injection is important and a benchmarking study can help better understand the vulnerabilities of LLMs to prompt injection. I appreciate the efforts of this paper in releasing the benchmark.

However, the benchmark can be further refined before publication. First, I agree with the reviewer gV14 that the researchers cannot practically use the entire dataset. I would suggest curating the dataset a bit, e.g., the paper could select a subset of or categorize samples that can easily be used to evaluate defenses. This is also mentioned by reviewers mg4k. Second, as noted by multiple reviewers, the evaluation is limited in scope. For example, the discussion of defenses in Section 2.2 is brief, and many existing defense techniques are neither included nor evaluated. Similarly, the set of LLMs considered in Section 2.3 is limited. As a result, not many interesting findings were made (e.g., it is already known to the community that LLMs are vulnerable and simple defenses would not work).  As a benchmarking paper, I would expect a more comprehensive evaluation in replay settings (this is also the setting in which this benchmark can be used by other researchers). For the above two reasons, I would suggest refining the paper and resubmitting it.

**Reviewer Scores:**

The reviewer gV14 would change the score to 6. The other three reviewers may maintain their scores.

---

### Decision · Program_Chairs · 2026-01-26

Reject